# FPGA Implementation of IEC 61131-3-Based Hardware-Aided Timers for Programmable Logic Controllers

Miroslaw Chmiel [1,*], Robert Czerwinski [1] and Andrzej Malcher [2]

1 Department of Digital Systems, Silesian University of Technology, 44-100 Gliwice, Poland; rczerwinski@polsl.pl
2 Department of Electronics, Electrical Engineering and Microelectronics, Silesian University of Technology, 44-100 Gliwice, Poland; amalcher@polsl.pl
* Correspondence: mchmiel@polsl.pl; Tel.: +48-32-237-1316

**Abstract:** Designs of timer function blocks (FBs) are presented in the article. The developed modules are IEC 61131-3. An analysis of IEC 61131-3 in terms of timer functionality and implementation options is presented. Three types are presented, timer-on, timer-off, and timer-pulse, with each type designed to be fully hardware or software-like. Both designs, hardware or software-like, can operate as multi-channel timers. Particularly noteworthy is the software-like design, for which a solution without edge detectors was achieved. Such a feature was obtained by reversing the method of time determination by counting the difference between the start and end times and by using specific features of the D flip-flops, that is, clock-enable inputs. The presented timers were written in Verilog language and implemented in an FPGA chip. Thanks to the universal design of the interface, the proposed FBs can be used for the hardware support of existing programmable logic controllers (PLCs) or as an integral part of newly built PLC CPUs. The idea of a CPU architecture with hardware support is proposed. The paper presents the results of the implementation in an FPGA of the Kintex UltraScale+ family from AMD-Xilinx.

**Keywords:** IEC-61131-3; timers; field programmable gate array; function block; programmable logic controller

## 1. Introduction

IEC 61131 is an international standard for programmable logic controllers (PLCs). IEC 61131-3 defines languages and software architecture [1]. It mainly defines text and graphics languages. Necessary elements, such as data types, variables, function blocks (FBs), and others, are defined. In this paper, the instruction list (IL) language is of particular interest. IL is an assembly-type language, originally created to be easily implemented in dedicated PLC execution units, as well as in units based on classic microprocessor structures. Graphical and textual program representations can be converted to IL in most implementations. It can be hypothesized that a microprocessor should be optimized for the efficient implementation of IL code. Unfortunately, the standard is not precise. It requires many amendments and comments [2–4]. Nevertheless, the provisions of the standard are implemented in industrial PLCs and in experimental or academic structures.

Field programmable gate array (FPGA) technology in particular plays a big role nowadays. FPGAs are customer-configurable integrated circuits. The main structure contains programmable interconnects and logic, allowing the flexible implementation of designed functions. This makes it possible to design application-specific hardware circuits. In addition, microprocessors can be implemented in FPGAs as hard or soft cores, so a system-on-a-chip could be built [5–7]. Recently, it has become increasingly common for soft-core microprocessors to be based on the ISA RISC-V specification [8]. The RISC-V core makes it possible to efficiently perform operations especially on data contained in working registers. Achieving high speeds of the central unit is closely related to the expectations of

Industry 4.0. To this end, dedicated microprocessors are now being developed to enable not only control program execution but high-speed IoT communication [9].

A lot of effort and time is required to design a dedicated microprocessor. So, it is much simpler to design a PLC using a microcontroller/microprocessor off the shelf [10]. Another tool that allows standard programming languages to be used for PLCs is a program industry safety translator into ANSI C language [11]. The resources of a classical microprocessor are not aligned with the IEC 61131-3 standard. By developing a specialized microprocessor, these drawbacks can be eliminated [7,12–14].

The logic structure allows the implementation of true concurrency, which is a major advantage over microprocessors. Improved PLC design using FPGA has been proposed based on parallel execution mechanism for enhancement of performance and flexibility [15]. Also, it is possible to implement the control program directly into the logic circuit. Such an opportunity is provided by FPGA technology. In the papers [16–19], the program is converted to a hardware description language. Next, such a description is synthesized. However, this new approach is still being developed. It does, however, take advantage of the ability of FPGA to run multiple components concurrently.

The concurrency of a programmable devices can also be used in a completely different way. The idea is to have a high level of abstraction of the circuit description (e.g., Matlab or HLS tools) and then perform automatic logic synthesis [20,21]. Such an approach, although still inefficient in terms of the synthesized structure, is also gaining adherence in the case of PLC design.

Finally, an inherent feature of FPGAs is reprogrammability, which was also reflected in the design of the PLC [22].

According to IEC 61131-3, an FB is a software organizational unit that, when executed, can provide one or more values to its outputs. FBs can be used multiple times with a unique data structure that stores information about the state of the called FB (static variables). Classically, FBs are implemented as software procedures, which is their primary drawback. Their execution requires a lot of time. Therefore, blocks with special capabilities, such as high-speed counters (HSCs), are designed [23]. Hardware implementation even makes it possible to build a classical counter in such a way that it behaves like a fast counter [24].

The present work deals with the implementation of one of the basic blocks of standard FB (SFB; [1])—timers. In industrial real-time systems, timing circuits play a key role alongside combinational logic and memory circuits. The most common applications of timing systems in automation systems are as follows:

- Switching an actuator on for a specified period of time;
- Switching an actuator on or off with a specified delay relative to a detected event;
- Supervision of the correctness of the operation of automation devices (an error is generated if there is no confirmation of operation within the assumed time from the issuance of the control signal).

The main contribution of the paper is to propose a structure of the timer FBs for PLCs compliant with the IEC 61131-3 standard. All three designs are developed and presented, timer-on, timer-off, and timer-pulse, with each type designed to be fully hardware or software-like, and able to operate in multi-channel mode. The idea of a CPU architecture with hardware support is proposed. Finally, the paper presents the results of the implementation in an FPGA of the Kintex UltraScale+ family from AMD-Xilinx.

## 2. The Timer

Figure 1 shows a graphical representation of the timer. The interface of such FB is defined by the following:

- IN (input) [BOOL]—triggering input; the edge of this input initiates the time measurement;
- PT (preset time) [TIME]—value for evaluation of the Q state;
- Q [BOOL]—status of the timer;
- ET (elapsed time) [TIME]—value of actual state of the timer;

- <TYPE>—TON (time ON-delay time), TOF (time OFF-delay time), or TP (pulse timer) timer function;
- <T_Name>—name of the timer instance.

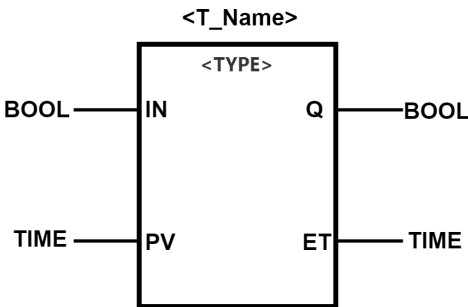

**Figure 1.** Interface of the timer in Ladder Diagram language.

The declaration of the inputs and outputs of the timer TP function module within the Structured Text language is described in the standard [1] and is shown in Listing 1.

**Listing 1.** Declaration of the TP timer in Structured Text language.

```
FUNCTION_BLOCK TP
IN (BOOL)
PT (TIME)
Q (BOOL)
ET (TIME)

VAR_INPUT
  IN : BOOL;
  PT : TIME;
 END_VAR

VAR_OUTPUT
  Q : BOOL ;
  ET : TIME;
END_VAR
```

The implementation of timers needs mapping of a data structure in the PLC memory that store the current content of the timer ET, the status of the PT input and the status of Q output. However, Q and ET can be calculated at the moment of reading, so the data structure may not contain them. An additional memory cell may be necessary to store previous statuses of the IN (IN'). The actual set of memory cells associated with a timer is closely related to the implementation used in a particular PLC solution.

Any attempt to describe the work of the timers should, however, begin with an analysis of the time charts presented in Figure 2.

The standard defines three types of timers:

- TON—when IN is active, the timer counts time up, and after reaching PT, the binary output Q is activated. Output ET shows the actual time state of the timer.
- TP—gives a pulse on the output after positive edge detection on the input.
- TOF—when IN is becoming inactive, the timer counts time up, and after reaching PT, the binary output Q is deactivated.

Pseudo-code that explains the operation of the different types of timers is presented in Listing 2.

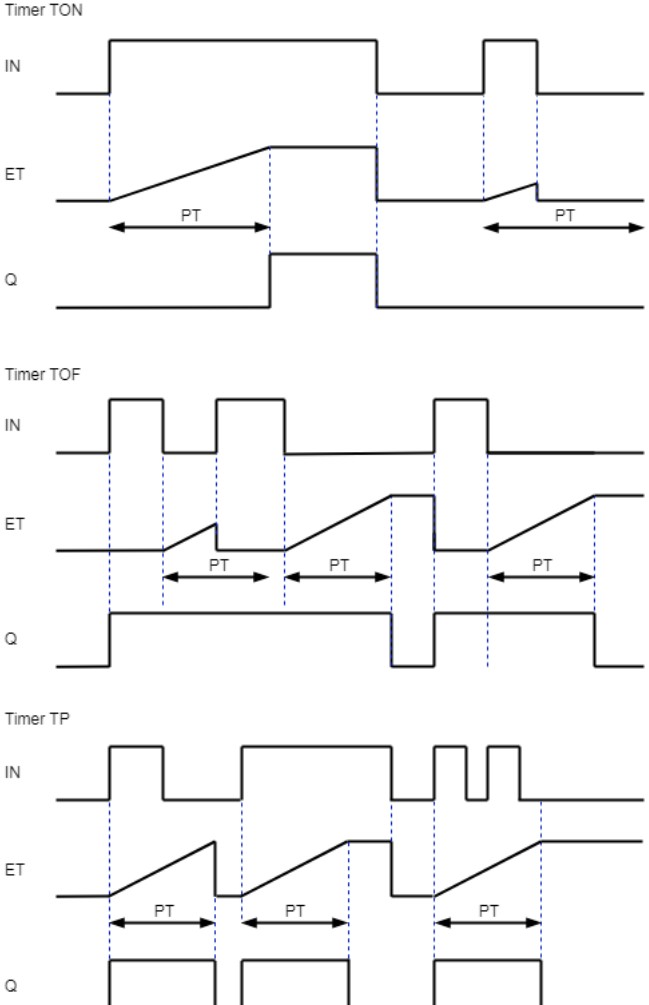

**Figure 2.** Timing diagrams for IEC-61131-3 timers.

**Listing 2.** Description of the timer presented in the standard.

```
//Time On Delay TON

IF !IN THEN ET := 0;
ET_EN := IN * (ET<PT);

Q:=IN * (ET>=PT)

//Time OFF Delay TOF

IF IN_FP THEN ET := 0;
IF IN_FN THEN IN_INT := 1;
IF IN_FP | (ET>=PT) THEN IN_INT := 0;

ET_EN := IN_INT;

Q := IN | INT_INT;

//Pulse Time TP

IF IN_FP THEN IN_INT := 1;
IF !IN * (ET>=PT) THEN IN_INT := 0;
```

```
ET_EN := IN_INT * (ET<=PT);
IF !IN_INT THEN ET := 0;
Q := ET_EN;
```

### 2.1. The Timer Operation

The description of timers in the standard is laconic. It mainly focuses on showing the time waveforms during their operation. In view of this, any designer of a PLC CPU will have to answer some basic questions:

1. With what accuracy should timers measure time?
2. What should be the range of measured time?
3. How and where is the type of timer determined, that is, how does the timer measure time? To be more specific: when does the timekeeping process start, under what conditions does it continue, when does it end, and what is the generation of the binary output state of the timer Q?
4. How and when does the controller store the state of the inputs/outputs in the context of each timer update?
5. How and when does the timer inform the control program of changes in its structure, that is, when are the states of the timer memory cells modified (by which the program recognizes that something has changed in the timer)?

Regarding the first two questions, the standard suggests a solution. This is because it defines two types, TIME and LTIME:

- TIME—the range of values and precision of presentation in these data types is implementer specific.
- LTIME—the data time LTIME is a signed 64-bit integer with units of nanoseconds. The update accuracy of the values of this time format is implementer specific (i.e., the value is given in nanoseconds, but it may be updated every microsecond or millisecond).

The selection of values and precision can be determined using previous experience and what has been implemented in existing controllers on the market. From the analysis, it appears that most of the timers offered in commercially available controllers measure time with a resolution of 1 ms, and the range of this measurement is an integer, which is stored as 32 bits (the DINT format). This gives the possibility to measure $4.29 \times 10^9$ ms, which is less than 50 days, which seems to be completely sufficient for most industrial facilities. In contrast, for the LTIME format, we obtain $18.45 \times 10^{18}$ ns, which gives 213,504 days.

As for the type of a particular timer instance, it is assumed that this is defined in the declaration part of the program, but the standard is quite poor in the details. In graphical languages, there is an explicit designation of the timer type, which is not seen in textual languages.

Crucial to the implementation of the timer project is the answer to the last two questions. The answer is quite complex, and this topic is discussed in the next section.

### 2.2. The Call to the Function That Triggers the Timer

The standard defines three ways of calling timer operations in IL. The first is to call an FB with the parameters passed in parentheses [1], as presented in Listing 3.

**Listing 3.** The timer CAL.

```
CAL TMR (IN := IN\_BOOL , PT := IN\_TIME, Q => BOOL\_OUT, ET => TIME\_OUT)
```

This type of call may not contain a complete list of assigned parameters but only the selected parameters. In such a case, the unassigned parameters take the values stored previously ([1]: see Table 42 on page 105). This way of calling is available in Concept [25] or CoDeSys [26] software, as well as in the SIMATIC S7 IEC timers [27].

The second way consists of preparing suitable data in particular fields (using LD and ST instructions) of the timer structure and then calling the function, updating the timer without any parameters using the CAL command. If not all fields of the structure are

assigned, the function uses the values of the structure fields stored earlier. As shown in Listing 4, there is a set of instructions (LD instruction) for loading data into the current result (CR) register and storing data to the individual fields that make up the timer structure (ST instruction). As opposed to counters, where functionality is realized only when the CAL instruction is executed, a timer counts the time units, so it must work all the time. Timer handling consists of two sets of actions: trigger handling (based on the IN-input state and the current timer state) and timer refreshing, which may or may not be performed as part of the CAL execution. There are four moments when the timer has to/may give new results—is refreshed:

- After execution of CAL;
- After each base time is reached;
- When the timer is read;
- At the end of each program loop.

This method is available in the Concept [25], CoDeSys [26], UnityPro [28], PLCopen [29], and ISaGRAF [30] environments.

**Listing 4.** Timer block calling using the CAL instruction.

```
LD INP                   //INP -> CR
ST TMR.IN                //CR -> TMR.IN
LD IN_PT                 //PT -> CR
ST TMR.PT                //CR -> TMR.PT
CAL TMR          //Timer Block Call (Execution)
LD TMR.Q                 //TMR.Q -> CR
ST OUT_Q                 //CR  -> OUT_Q
LD TMR.ET                //TMR.ET -> CR
ST OUT_ET                //CR  -> OUT_ET
```

However, the provisions of the standard also allow the use of so-called short IL operators. The problem is how to refresh the timer while calling short operators IN and PT (Figure 3). Regarding the short operators, the standard is somehow weak. Each short operand updates the timer state, so it improves the conciseness and program readability. But an analysis of the standard leads to the conclusion that each use of the short command is followed by CAL. This significantly affects the implementation of the timer block and has a big impact on the execution time of this block. In the Siemens SIMATIC S7-300/400 PLCs, the basic timers do not form an FB. Therefore, there is no problem using only timer elements that are necessary at that moment. It may seem that in this case the processing time of the timer is reduced. Among the implementations of the standard known to the authors, only Concept allows the use of operator-based programming. UnityPro, CoDeSys, and ISaGRAF do not allow this method of timer calling. An example code is presented in Listing 5. A program written using the short operators needs to be translated into short operators with an implicit CAL instruction. As a result, it takes longer to execute than a program explicitly using the CAL command.

**Listing 5.** Timer block in IL using operators.

```
LD         IN_PT
PT         TMR     ;Equivalent to: ST TMR.PT + CAL TMR
LD         INP
IN         TMR     ;Equivalent to: ST TMR.IN + CAL TMR
;Operators take into account the current state of all bits in the TMR structure
;Explicit CAL not required
```

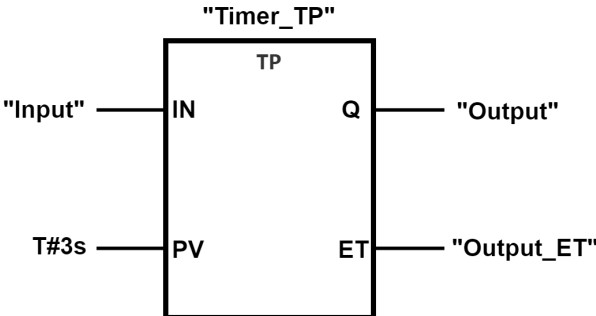

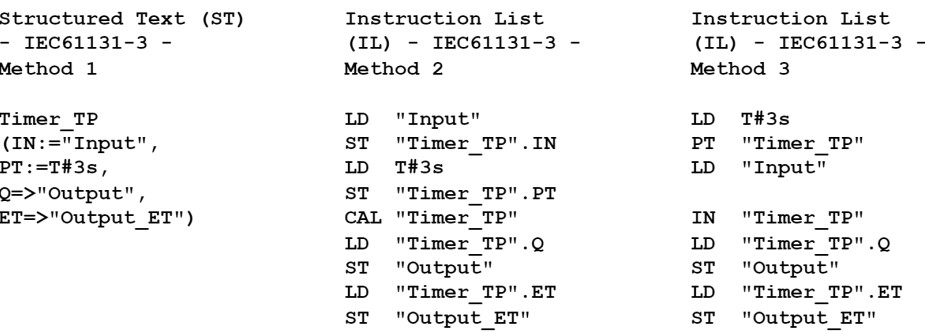

**Figure 3.** Methods of calling the timer's block for IEC-61131-3.

### 2.3. Ways to Measure Time

The concept of measuring time can have two meanings. The first meaning is the action of determining with satisfactory accuracy how long a certain event lasted. The result of such an action is a number expressed in units of time (usually seconds with a fractional part) giving the measured duration of the event. The second meaning of the concept of timing is to generate a binary sequence (pulse or delay) of a given duration. In this case, the control quantity is the number of time units, and the result is a waveform at a specific output in response to a so-called trigger event. Timers used in industrial automation systems realize timekeeping in the latter sense. However, it is also possible to use timers to determine the duration of an event occurring at a specific input, but this is not among their primary uses.

Ways to implement timers in PLCs include the following:

1.  Analog timers—these types of timers use integrated univibrators with an external resistor and capacitor circuit. The pulse duration is set in the hardware, using a potentiometer. This type of timer can be found in companies' early industrial controller designs. The disadvantage of such a solution is the low accuracy and repeatability of the pulse length, as well as the inability to change the setting by program.
2.  Counter-based timers—the principle of operation is based on counting pulses of a reference frequency. The resolution of these timers is determined by the reference frequency. Virtually all timing systems encountered today belong to the group of counter timers. The systems differ significantly in the way they refresh (i.e., update the time value) in the timer register. The refreshing methods encountered can be divided as follows:

    (a)  Timers refreshed cyclically by the operating system of the controller. In this type of timer, refreshing occurs at every specified interval, such as using a cyclic interrupt. In each cycle, the state of all timers is refreshed regardless of whether they are used. This requires hardware support. The data structures for the timers usually occupy a specific place in memory, and there is a strictly defined number of them. The frequent refreshing of all timers takes a considerable amount of time, which can lead to a noticeable slowdown in the speed of the

controller. To avoid slowing down the operation of the CPU, the timers refresh every 10 ms or less frequently, which leads to a lower resolution of timekeeping.

(b)　Timers refreshed during the execution of a timer instruction. This method requires the timer instruction or timer-triggering function to be designed so that each time a program segment containing this instruction is executed, the timer state is refreshed. The advantage of such a solution is that the handling functions do not slow down the controller cycle (there are no cyclic interrupts designed to refresh all active timers). Also, the number of timers running in this way is not limited by hardware and depends only on the size of available memory. Triggering a timer operating on this principle involves rewriting the state of the system millisecond counter to the timer's "Start" register. Refreshing involves subtracting the value stored in the "Start" register from the current value of the system millisecond counter and writing the difference to the ET register. This results in a rather significant disadvantage of this type of timer, which is the need to store as many as three numerical values: the initial value of "Start", the preset value of PT, and the currently measured value of ET. In modern control systems, where memory capacities are much larger, this disadvantage has lost its importance.

(c)　Timers refreshed every controller cycle. PLCs work in a serial-cyclic manner, that is, after each execution of the main block, operating system activities are carried out. These activities include rewriting the output image memory to physical outputs, rewriting the status of physical inputs to the input image memory, handling communication tasks, diagnostic functions, and so on. It is possible to include timer refresh activities in this phase of the controller cycle. This allows for a significant simplification of the activities performed by the program during the execution of the segment containing the timer instruction—it is only required to rewrite the state of the reference counter (millisecond counter) to the "Start" register of the timer. All other activities are handled systemically at the end of the cycle. The memory requirements of such a solution are similar to those of solution 2b, and the time outlay is greater and constant, because all timers are required to be refreshed after each cycle regardless of whether they are used in the program. The resolution of the timekeeping is limited by the cycle period of the controller, and this depends on the size of the program.

(d)　Timers refreshed at the time of reading. This method of refreshing the timer is based on the observation that the state of the timer has no meaning until that state is checked. Thus, the refresh of the timer is performed immediately before reading its state in the form of a Q bit or ET value. The memory requirements for such a solution are identical to those of cases 2b and 2c. The complexity of the triggering instruction is analogous to case 2c. The advantage is the availability of the most recent Q and ET value at each reading, even if the reading is performed several times in a program cycle. The timing resolution does not depend on the program size and cycle time—it is always equal to the period of the reference counter (usually 1 ms).

Refreshing at the beginning of the cycle and during the execution of the CAL command means that the accuracy of the measured time reading is limited by the cycle time of the software loop. Asynchronous (system) refreshing and refreshing at the time of reading guarantees that the read time value is always up to date. Therefore, when designing a timer block operating in an FPGA chip, the (b) and (c) refresh methods were rejected.

## 3. The PLC Architecture

In order to design a timer FB, it is necessary to propose the architecture of the central processing unit and the interface of the designed FB.

### 3.1. The bit.WORD IHSPLC

A classical PLC is equipped with functions and procedures that are responsible for the functionality of FBs. This solution is based on software processing (Figure 4). Program procedures for standard FBs are part of the PLC operating system. The execution of tasks involves creating the appropriate data structure in the data memory and executing the specified function. It is necessary to allocate memory for data structures, and worse, to devote CPU time to call the specified function. Such a unit turns out to be inefficient from the point of view of resource consumption and the execution time of individual tasks. FPGAs provide opportunity for hardware support of some FBs tasks.

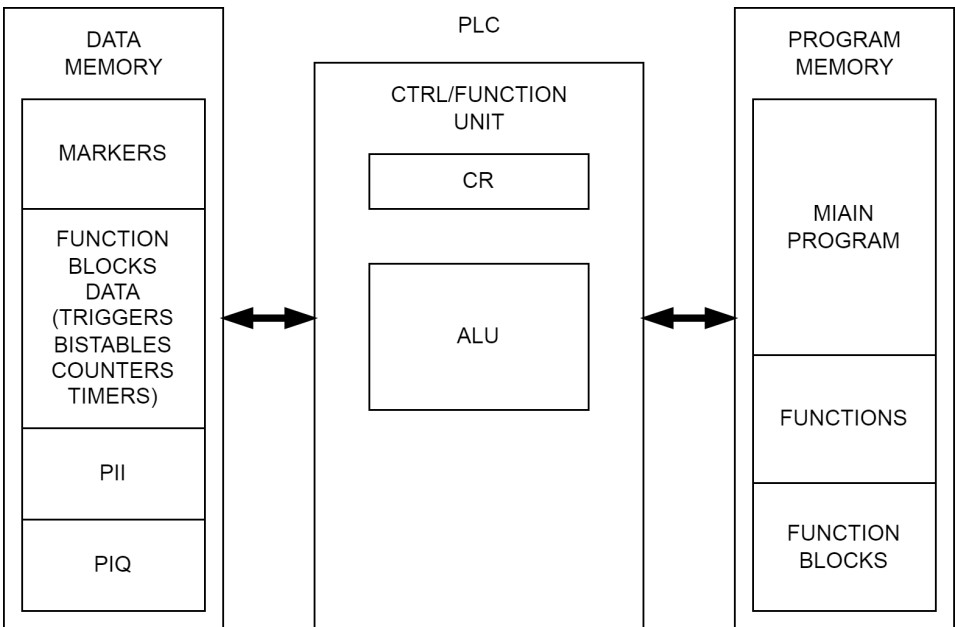

**Figure 4.** Architecture of a classical CPU.

The solution proposed in the paper is to construct a controller with integrated blocks of hardware that support the performed operations in the integrated hardware–software PLC (IHSPLC). Moreover, a bit.WORD unit with a CPU consisting of two parts is shown. It can be implemented in the FPGA device. Figure 5 shows the controller implemented in this way.

The basis of the integrated hardware–software controller is a dedicated CPU with a developed microprocessor, the machine language of which complies with the IEC 61131-3 standard. Problems of the CPU construction are presented in [7], but this concept is combined with the bit-byte idea presented in [31–33]. Basic operations and binary-based FBs are performed in one clock period—complex operations take the same amount of time as the basic instructions [24]. The bit.WORD IHSPLC proposed in this paper works on bit and word CRs accumulators (CR_b and CR_W). The compiler decides, based on given the data type, which CR should be used.

### 3.2. The Timer Interface

To accelerate the timer operation, appropriate logic resources are used in FPGA. So, the functions that were responsible for processing data are replaced with hardware logic blocks that connect data structures with operations. However, the main problem concerns the interface between CPU and the hardware units.

For soft microprocessors, FBs can be integrated into the CPU structure. However, the hardware accelerated block must be designed with minimum latency for communication to ensure a minimum delay for CPU-FBs data exchange. Therefore, the structure of timers FB proposed in the paper is driven only by means of the following (Figure 6):

- The CR—CR_b and CR_W;
- Address—TMR_ADDR;
- Type and enable signals—TMR_TYPE_CTRL;
- Value of actual state of the timer—TMR_ET;
- Status of the timer—TMR_Q.

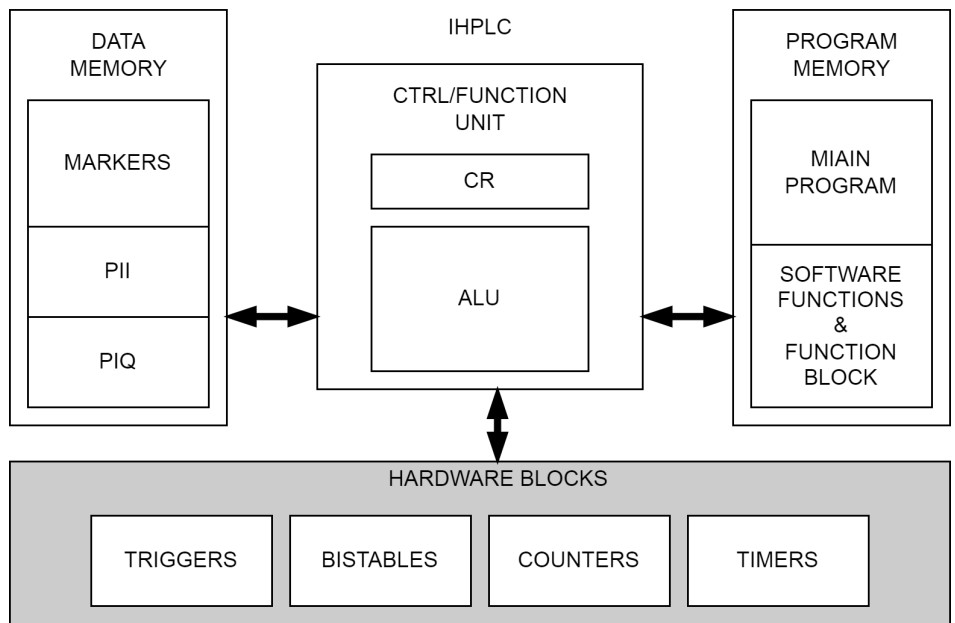

**Figure 5.** Structure of the bit.WORD IHSPLC.

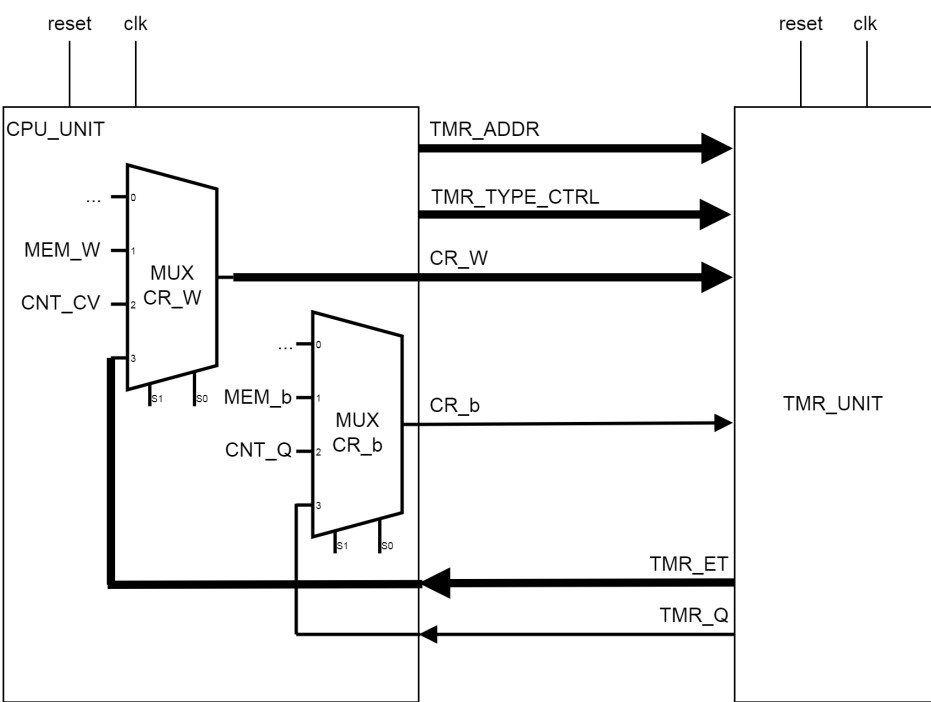

**Figure 6.** Interface between the hardware-supported block and function unit.

The designed timer block also has access to the system clock and reset. The timer block (TMR_UNIT) returns TMR_ET and TMR_Q. However, the CPU expects these signals to be connected only to CR_b and CR_W. Therefore, it is necessary to multiplex the signals TMR_ET and TMR_Q appropriately with CR_W and CR_b as shown by the CPU_UNIT block in Figure 6.

## 4. FPGA Timers Implementations

The timers' FB is proposed in this section: hardware and software-like timers are presented.

### 4.1. Hardware Timers

One of the most important problems in the design of a timer block is how to update the timer, that is, to determine the moment of execution/actualization. This is, of course, closely related to the way such a block works. Let us consider a classical binary counter, which is stimulated by a reference signal. At the output of the counter will be a value adequate to the number of pulses, which corresponds to the measured time. This value will be updated in real time, which follows from the principle of operation of the presented scheme. To ensure functionality in accordance with IEC 61131-3, it is necessary to add several elements to this simple scheme. Figure 7 shows the structure of the hardware counter-based timer. The timer is updated based on the state of inputs such as 'IN' (CR_b) and 'PT' (CR_W).

The timer can be driven only by means of CR_b, so those inputs must be stored somehow. Moreover, one of the most important parts of the timer is PT, which must be remembered. Figure 7 shows the structure of the TON timer. The basic element of the system is a counter that counts clock pulses with a frequency of 1 kHz (1 ms), when the signal EN = 1. The conditions of the activity of this signal are shown in Listing 2. The counter in the situations specified in the same listing is reset. It has a built-in comparator to detect the situation when ET >= PT. This comparator co-creates, together with the stored state of the IN input, the state of the 'Q' output. Its elapsed time is derived as 'ET' output straight from the counter.

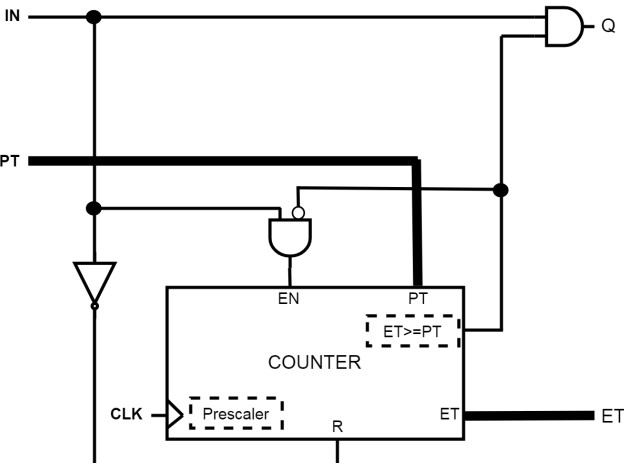

**Figure 7.** Structure of the hardware-based TON timer.

TOF and TP timers can be developed in a similar way as for the TON timer shown in Figure 7. However, they require a different, slightly more elaborate structure modeled on the entries in Listing 2. The structure of a universal timer is shown in Figure 8. The timer's input must be remembered, but the timer may be executed after the CAL instruction. CAL instruction is realized by means of clock-enable (CE), which is input in the FPGA D flip-flop.

In the case of multi-channel implementation, it is necessary to place multiple timers in a single structure. By using the structure shown in Figure 8, the inputs are associated with the corresponding addresses because the enable inputs (EN) are active only for a particular address. The outputs must be multiplexed, with the output multiplexers (for ET and Q) being addressed from the address input as shown in Figure 9.

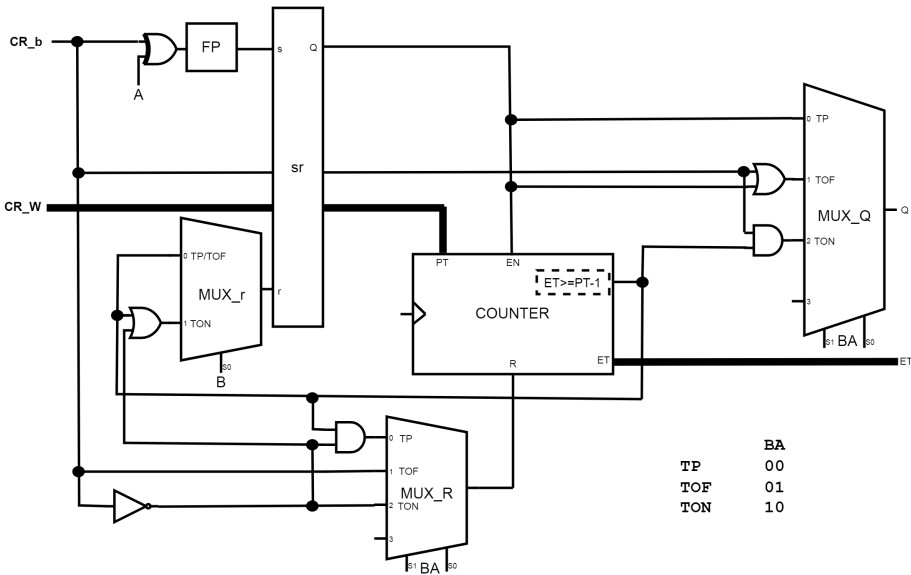

**Figure 8.** Structure of the universal timer: hardware-based timer.

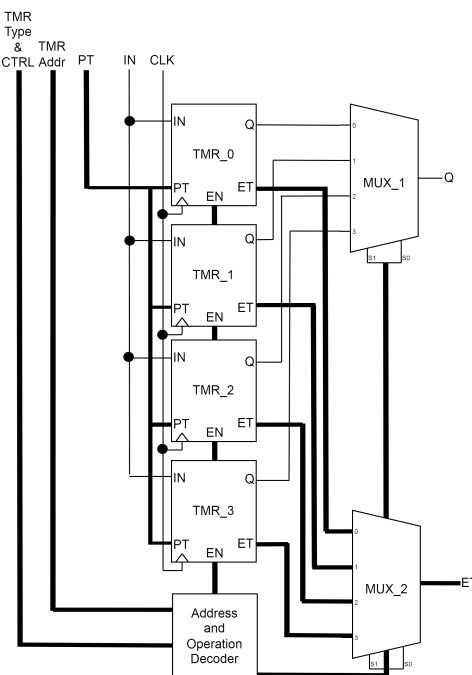

**Figure 9.** Structure of the timer FB.

## 4.2. IN and PT Memory Block

Each timer block in the figure requires the use of memory that allows storage of the state of both current registers, both CR_b and CR_W. The stored states of the condition registers in the block are used as IN and PT signals. The design of the IN and PT memory is shown in Figure 10. To ensure the possibility of using the second and third methods from Figure 3, it was additionally necessary to ensure that saving variables was possible in two ways—using the CAL command directly existing in the program and without this instruction. Method 2 (Figure 3) always requires the use of two levels of memory—the first level is used to store the state of the appropriate signal during the execution of the appropriate ST operation, while the second level is used during the execution of the CAL operation. Then, the new IN and PT values affect the internal circuit of the timer. Method 3, assuming there is also a CAL in the IN instruction, requires only one level of memory

to be used for the IN variable—the IN instruction copies the state of CR_b directly into IN memory. The same way of saving the states of CR_b and CR_W must be used when implementing the timers described in the next chapter. However, in these solutions, we are dealing with memory blocks, as shown in Figure 11.

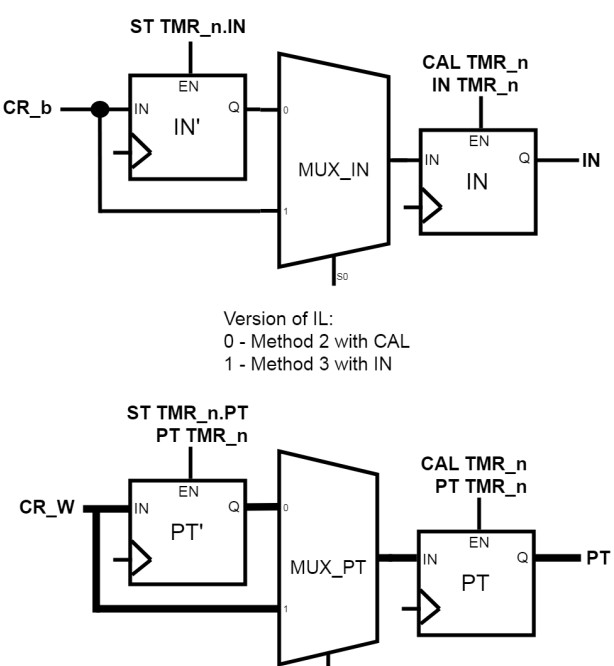

**Figure 10.** Construction of an input block for the hardware timer block.

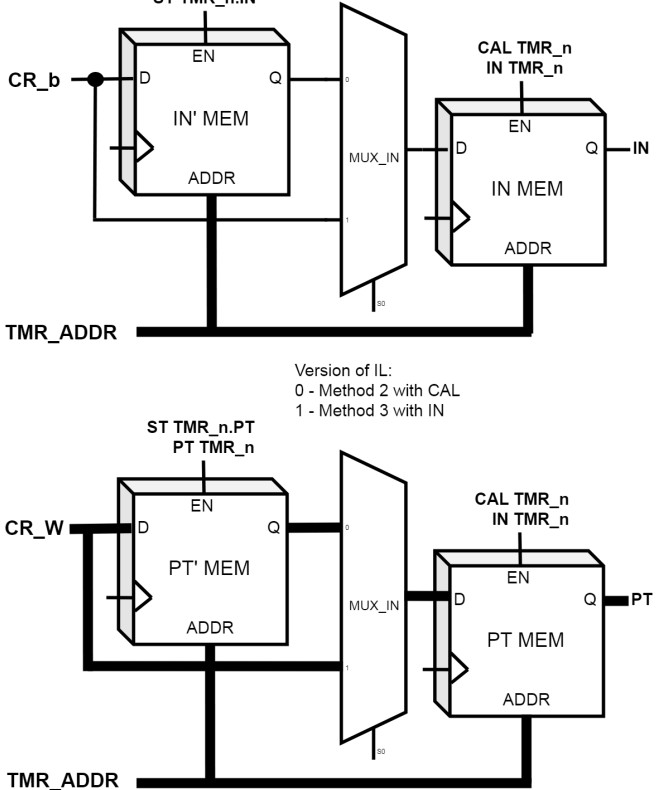

**Figure 11.** Construction of an input block for the software-like timer block.

As already mentioned, the IEC standard remains ambiguous in many places and allows for different interpretations of particular concepts and resources depending on the implementation. Siemens, in the IEC timers of both the S7-300/400 and S7-1200/1500 PLCs, decided to realize only one PT memory. Although a compliant IL language has not been implemented, a PT command (equivalent to the PT operator) has been introduced in the Ladder Diagram and Function Block Diagram graphics languages. According to the [27] documentation, this command causes an immediate change in the contents of the PT memory, and thus the possibility for an immediate change in the state of the binary output Q. After the PT command, the CAL command is no longer executed. The advantages of implementing timers with a single PT stage are the efficiency of timer task execution and memory savings. However, the solutions presented in the article (Figures 10 and 11) are fully universal and, if necessary, memory cells and multiplexers in the PT path can be removed.

*4.3. Software-like Timers*

A software timer is just a variable that is incremented by particular events. Hardware-based timers, which are designed by means of D flip-flops are also well known. They work very fast in relation to software timers. Software timers must be executed somehow by means of the program (Listing 2, Section 2). A very big block of independent software timers can be built; however, those timers cannot work concurrently. This is not a problem in the PLC because it works serially by definition. On the other hand, there is serious drawback of a hardware-based big timer structure—the flip-flop utilization. So, the idea is to compound those two implementations—to implement the software idea by means of hardware.

To design software-like timers, memories, driven by means of CRs (CR_b and CR_W), are necessary. Each memory use a common address input. But it would not be time effective to read data to CR, perform an operation, and write back data to the memory. The best way is to design multiplexers that are driven by decoded instruction.

One can imagine a multi-timer structure that uses the structures with counters as shown in Figures 7 and 8. However, this would require a separate 32-bit counter for each timer instance, as well as a set of multiplexers and appropriate memories to store the necessary elements: CR_b, CR_W, PT, and edge detector memories. The authors present a timer concept that does not use many counters but only one counter for the entire CPU run, which eliminates the need to perform edge detection operations. Instead, there is an additional memory that remembers the (RTC) counter state in a 32-bit memory cell for each timer instance, as well as one common combinational circuit that determines the difference between the current RTC state and the one saved during the initiation of the timer operation. In this way, such a system continuously calculates, for the selected timer instance, ET and the state of the output Q. In the presented solution, the edge detectors present earlier (Figures 7 and 8) have disappeared because the process of locking the start time value occurs exactly at the moment of the EN signal for the start time cell. For example, for the TON timer—in Figure 12—the RTC state is written continuously to the start time cell whenever IN (CR_b input is FALSE) and when the RTC and start time states are equal (i.e., their difference is 0—ET = 0 and Q = FALSE). At the moment, when CR_b becomes TRUE, the value in the register (memory cell) is latched, and during this exact moment, the RTC and start time difference is calculated, which determines the ET value on an ongoing basis. After reaching the PT value, for the TON timer, the TRUE state appears on the Q output. Similarly, taking into account the differences in operation, the TOF type timer was implemented—Figure 13. For the TP timer—Figure 14—as shown in Figure 8, an additional flip-flop had to be used. For this type of timer, the need to use edge detectors was also avoided.

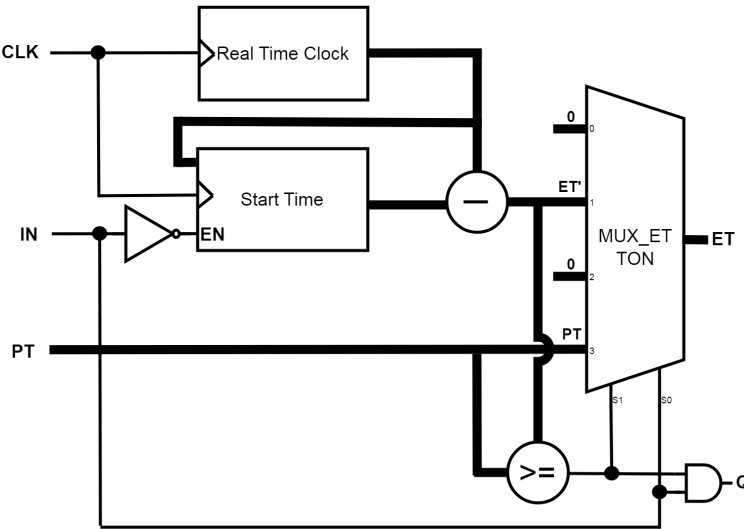

**Figure 12.** Structure of the software-like TON timer.

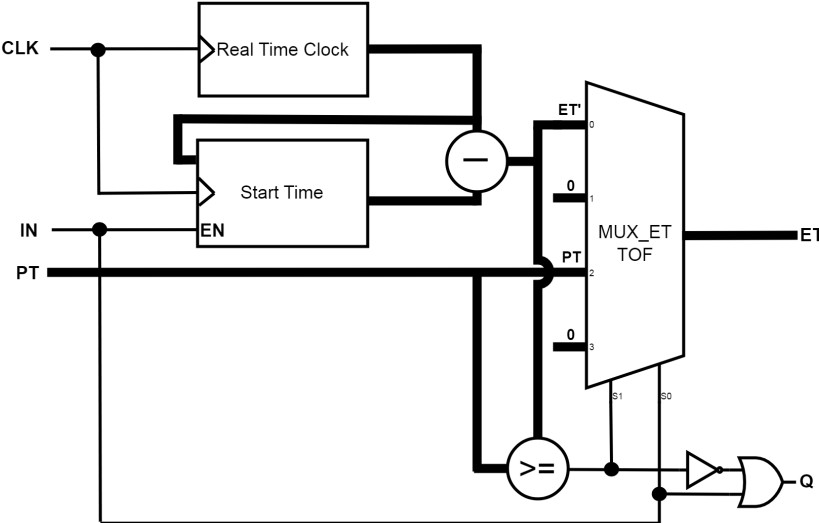

**Figure 13.** Structure of the software-like TOF timer.

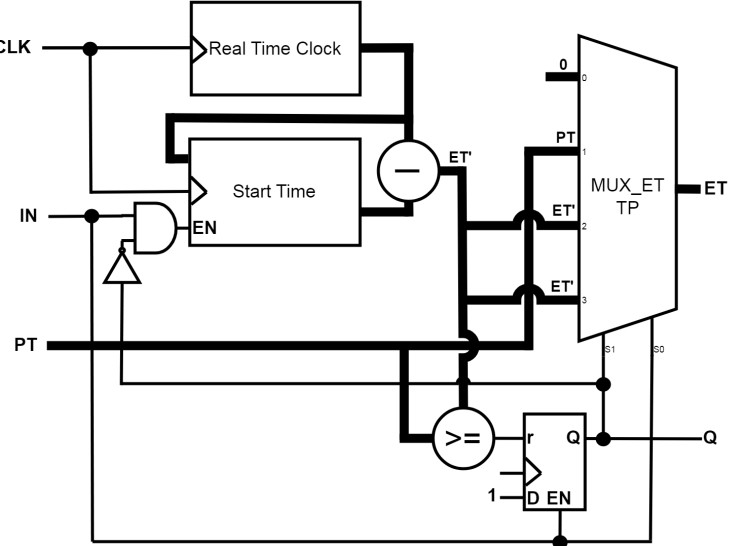

**Figure 14.** Structure of the software-like TP timer.

Figure 15 shows a multi-timer structure (called a multi-channel timer) that allows all types of timers to be implemented. The structure has all the necessary elements: one RTC counter and one combinational circuit calculating the difference with a built-in comparator (blue), three multiplexers (selecting the type of timer, realizing the state of the ET and Q outputs, and four memories), the state of inputs (IN cells), setpoints (PT), start values, and, only for TP-type timers, additional input status memory (TP). The system also has several logic gates necessary for resolving individual functions.

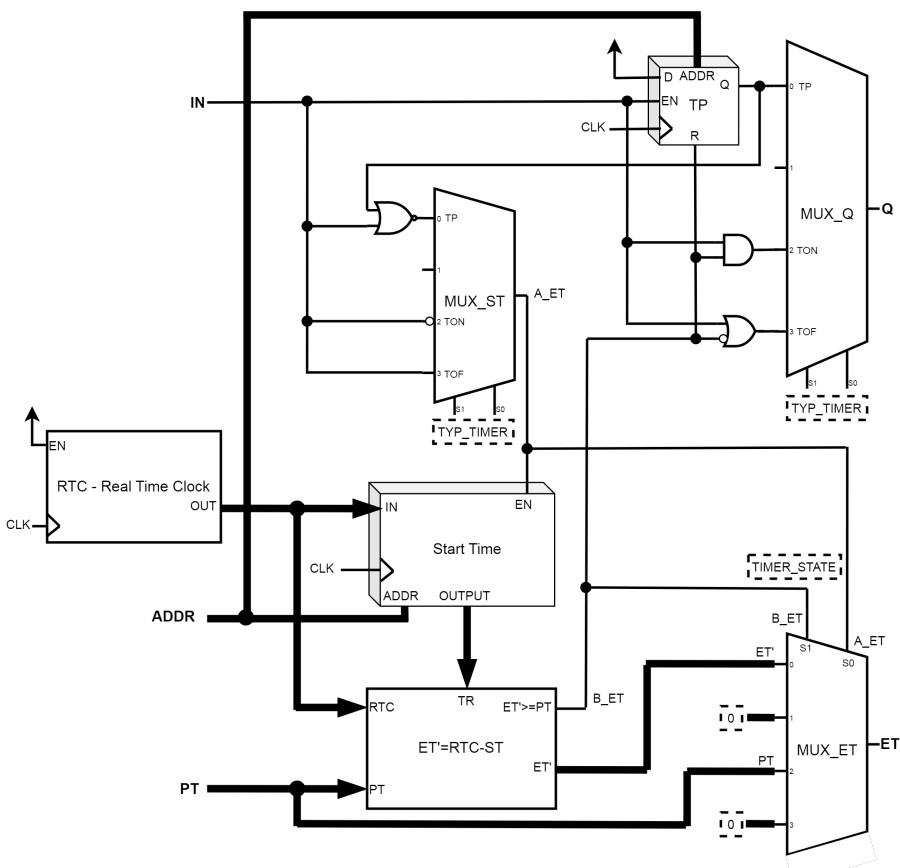

**Figure 15.** The software-like timer FB.

The question is whether binary outputs should be calculated continuously, determined during the execution of the command testing the binary state of the timer or executed during CAL execution. A fragment of a program for calculating the binary state of the timer using the basic operations of the controller is presented in Listing 6.

**Listing 6.** Proposed method of TON timer outputs generated using standard PLC operations.

```
// Q
LD RTC
SUB ET //ET'
GE PT //(GE - Greater or Equal)
ST TMR.Q
```

The implementation of such a program would require only that the computing unit has access to the ET and PT cells. The structure dedicated for short operators of a software-like timer is shown in Figure 15. The ET memory stores the current timer state, and the multiplexer on the data input allows the value to be written from the PT memory to keep the current value of the timer. The PT is modified by the state of the CR_W. The MUX_Q multiplexer is used to produce the state of the timer's binary output—Q, that is implemented in a combinational way. It is calculated after time expired and input

change. The Q output is calculated all the time (for a specified address). There is no need to implement the memory, and it takes no clock cycles to elaborate the Q value.

The presented in Section 4 the machine cycle is executed within one clock cycle—single clock edge solution.

## 5. Experimental Results

The software-like and hardware timers were implemented using Verilog HDL. Functional verification were run to search for bugs and to improve the designed constructions. The experimental results include two main comparisons:

- A comparison between units for FPGA resources utilization and maximum clock frequencies;
- A comparison of ready-made PLCs presented in the current paper solutions.

A comparison of the logic utilization for software-like and hardware timers is presented in Tables 1 and 2. The Xilinx Kintex UltraScale+ was used (xcku3p-ffva676-1-i). Synthesis was performed for different numbers of implemented timers: 16-1024 (address width: 4-bit to 10-bit).

**Table 1.** Direct comparison of the logic utilization for software-like and hardware timers.

| No. of Timers | Hardware Timers | | | Software-Like Timers | | |
|---|---|---|---|---|---|---|
| | LUT | LURAM | FF | LUT | LURAM | FF |
| 16 | 849 | - | 672 | 189 | 99 | 36 |
| 32 | 1730 | - | 1344 | 189 | 99 | 36 |
| 256 | 13,969 | - | 10,752 | 486 | 396 | 36 |
| 512 | 28,057 | - | 21,504 | 880 | 792 | 36 |
| 1024 | 56,116 | - | 43,008 | 1723 | 1584 | 36 |

**Table 2.** Comparison of the timers implementations in simple CPU.

| No. of Timers | Hardware Timers | | | | | Software-Like Timers | | | | |
|---|---|---|---|---|---|---|---|---|---|---|
| | LUT | LURAM | FF | BRAM | $f_{MAX}$ [MHz] | LUT | LURAM | FF | BRAM | $f_{MAX}$ [MHz] |
| 16 | 996 | 33 | 714 | 0.5 | 302 | 330 | 132 | 78 | 0.5 | 251 |
| 32 | 1855 | 33 | 1386 | 0.5 | 302 | 330 | 132 | 78 | 0.5 | 251 |
| 256 | 13,843 | 33 | 10,814 | 0.5 | 293 | 629 | 429 | 78 | 0.5 | 241 |
| 512 | 27,656 | 33 | 21,591 | 0.5 | 291 | 1025 | 825 | 78 | 0.5 | 236 |
| 1024 | 55,423 | 33 | 43,145 | 0.5 | 286 | 1877 | 1617 | 78 | 0.5 | 224 |

A direct comparison of the logic utilization for software-like and hardware timers is presented in Table 1 and Figure 16. The number of look-up tables (LUTs) used by the timers increases linearly with the number of timers, whereby hardware timers utilize only simple LUTs, and software-like timers utilize LUTs and LUTRAMs, which are consumed for memory construction. Note that the amount of resources consumed grows much faster for hardware timers than for software-like timers. In addition, hardware timers also consume flip-flops in their design, whereas the number of flip-flops in a software-like solution is constant.

Table 1 shows one deviation from the linear increase in the LUTRAMs used. For software-like timers, 16 and 32 timers use the same resources. This is because the LUTRAMs in the FPGA family are 6-input LUTRAMs, which means that for fewer than 32 timers, the LUTRAMs are used inefficiently.

Figure 16 shows the sum of LUT and LUTRAM resources due to the similar logical purpose of these blocks for building digital circuits. This makes it possible to make a close and direct comparison between the solutions.

To fully compare software-like timers with hardware ones, a timing analysis was performed. The developed timer units were implemented in a simple CPU. The purpose

of this experiment was to compare the consumed resources and the maximum frequency of the timer network in a fully functional unit. The idea is to implement CR_b and CR_W accumulators along with all multiplexers and memories (e.g., PII and PIQ) to enable the timer tasks. Such a comparison, first of all, gives a proper overview of the speed of the different types of timer blocks. The results of the implementation (resource utilization and maximum frequency) are shown in Table 2, whereas the comparison of the maximum frequency (in MHz) of the obtained units is shown in Figure 17.

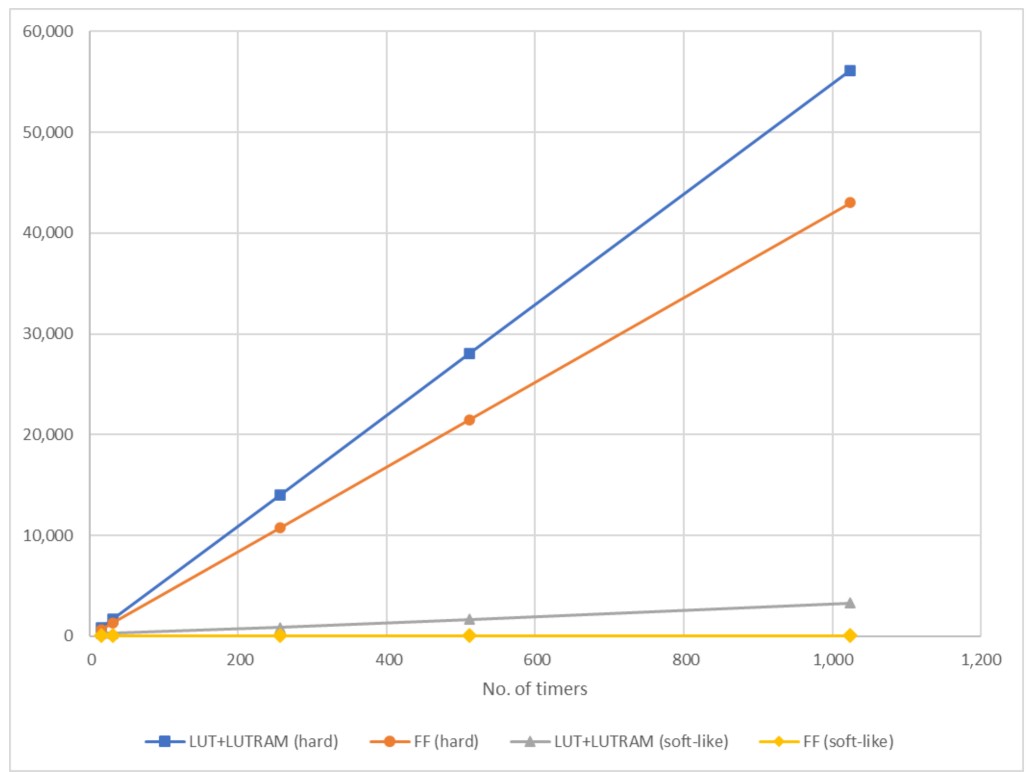

**Figure 16.** Direct comparison of the logic utilization for hardware and software-like timers.

Most significant in this comparison is the obtained maximum clock signal frequency—$f_{MAX}$. It is clear that the pure hardware solution is faster and yields frequencies of 282–302 MHz, whereas the analogous software-like solution yields frequencies of 225–251 MHz, which is about 17% lower compared to the hardware solution.

The results of the developed implementations were compared with the measurement results obtained for Siemens PLCs. An iterative algorithm was used to measure the execution time of timer commands, which was the basis for their comparison. The block diagram is shown in Figure 18. Before starting the next iteration, the current time is stored in the Start variable. For the assumed number of iterations (nc), the same measured instruction is called cyclically. At the end of the cycle, the instruction execution time is determined by the difference of the start and end times of the iterative process divided by the number of iterations. This method of measuring the instruction execution time is subject to error due to the iterative process itself, so in order to perform the calibration, the instruction under test was first removed from the program and the execution time of an empty program was measured. The actual instruction execution time was taken as the difference of the time obtained for the program containing the tested instruction minus the execution time of the empty program. To reliably determine the timing of the program, 100 measurements specified by the algorithm were taken. Coarse errors, such as those related to the operation of the operating system, were removed.

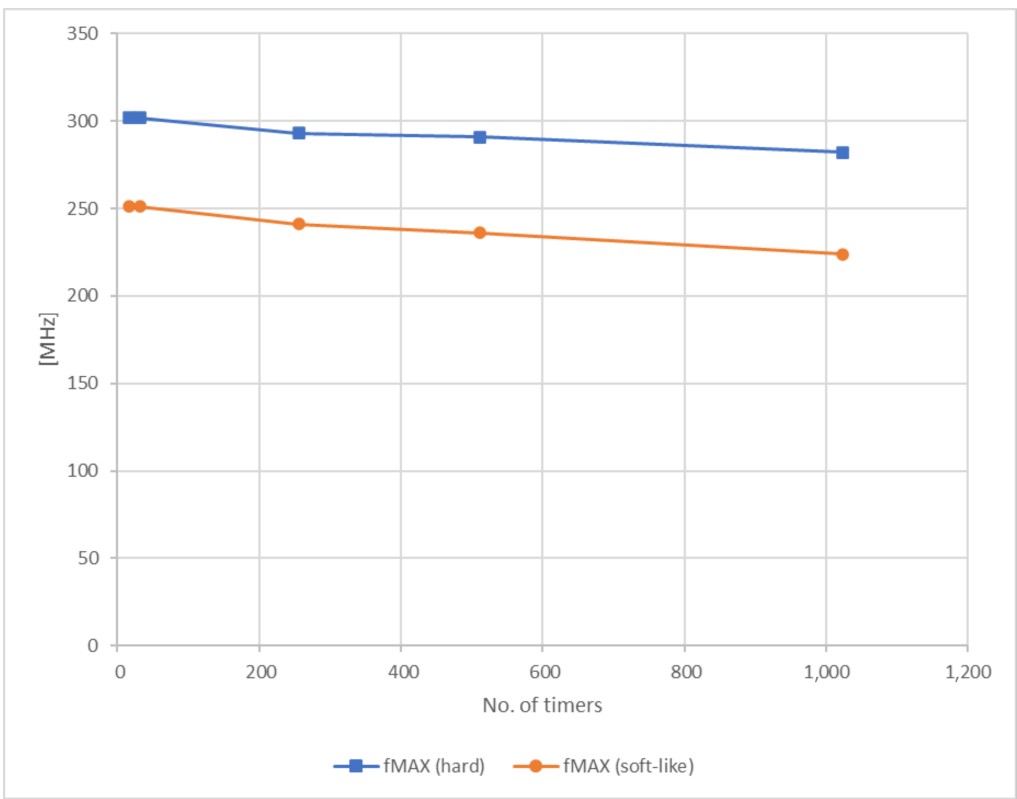

**Figure 17.** Direct comparison of the maximum frequency for the CPU with hardware and software-like timers.

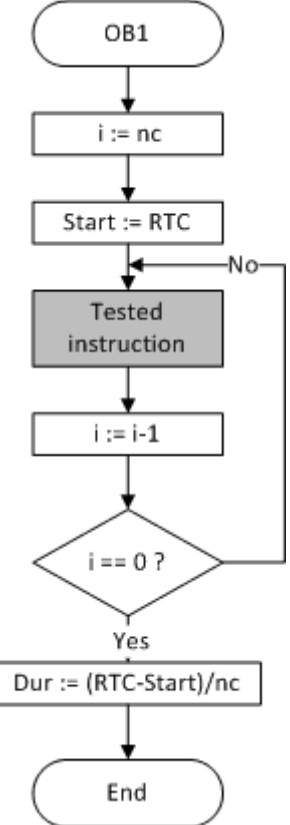

**Figure 18.** Block diagram of the algorithm used to measure the timing of timer commands.

A comparison of the time required to process all the timer operations (Listing 4) was made. The execution of each instruction requires exactly one clock cycle for presented CPU. The comparison was made with Siemens solutions—Table 3. The table shows the measured processing time for the program using all capabilities of the timer. For the controller S7-319 3PN/DP, there is huge disproportion between the implementation of the timer operations as a whole for the classical and the standard compliant solutions (Table 4). The structure proposed in the paper significantly exceeds the time performance of all other compared units. For example, in the S7-315 controller, the simple operation (A M x.y) takes almost nine times less than the state of the timer read instruction, and these operations for the S7-319 take almost four times more time to execute. Therefore, the execution times of individual timer tasks were compared with respect to the duration of the instruction that was assumed to be basic in every CPU controller: the instruction for reading the status of the contents of the PLC memory cell M (marker, memory). Thanks to this, it is clear that the implementation of timer operations proposed by the authors is much more time efficient than the implementation in Siemens PLCs, regardless of whether timers are 16-bit or 32-bit or whether the timers are the classic version (not compliant with the provisions of the standard) or the version compliant with the standard (most closely to short operators). In the best case, for the S7-315 controller and the timer not conforming to the standard, the difference reaches more than 10 times, whereas for the timer conforming to the standard and the S7-1516 controller, the difference is more than three times. The best result gives S7-319 3PN/DP for the non-standard solution of the timer—this CPU is only a little more than two times worse than the authors' solution. A very interesting example is the comparison of the execution time of the instruction for reading the binary state of the timer (output Q) with that for reading the bit state of the marker. For the presented controller, this quotient is, of course, 1, and the best result for the SIMATIC controllers is almost 2.5, a classic timer, and almost 4 (S7-319 3PN/DP), a timer compliant with the standard, and the worst is 19.25 (S7-1214). These differences are due to the fact that the timer operations for the proposed units, due to the hardware support, are implemented concurrently and take the same number of clock periods as the basic instructions (such as reading or writing a memory cell).

**Table 3.** Comparison of the execution time of the TON timers [ns].

| | S7-1214 | S7-1516 | S7-315 2PN/DP | S7-319 3PN/DP | S7-319 3PN/DP Classical Timer | This Work * Hardware | This Work ** Software-Like |
|---|---|---|---|---|---|---|---|
| Full TON timer block [ns] | 4460 | 335 | 21,500 | 1060 | 400 | 30 | 36 |
| Coil of TON timer [ns] | 1310 | 115 | n.a. | n.a. | 90 | 10 | 12 |
| Read ET cell (MOVE) [ns] | 1550 | 80 | 2000 | 80 | 250 | 10 | 12 |
| Read Q Cell (Switch+Coil) [ns] | 1510 | 115 | 1600 | 70 | 90 | 10 | 12 |
| Read ET and Q cells [ns] | 3100 | 210 | 3600 | 110 | 330 | 16,65 | 20 |
| Resolution [ms] | 1 | 1 | 1 | 1 | 10, 100, 1 [s], 10 [s] | 1 | 1 |
| Memory occupancy RAM [B] | 24 | 88 | 58 | 58 | 2 | *** | *** |
| ET data width [bit] | 32 | 32 | 32 | 32 | 12 in BCD | 32 | 32 |
| A M x.y [ns] | 40 | 10 | 220 | 19 | 19 | 3.33 | 4 |
| A Timer.Q [ns] | 770 | 90 | 1830 | 74 | 46 | 3.33 | 4 |

n.a.—non-applicable; * [300 MHz—3.33 ns/op.]; ** [250 MHz—4 ns/op.]; *** a direct comparison is inadequate due to different implementation technologies; please see Table 1.

**Table 4.** Comparison of the execution time of the TON type timer function versus the execution time of the basic instruction A M x.y.

| | S7-1214 | S7-1516 | S7-315 2PN/DP | S7-319 3PN/DP | S7-319 3PN/DP Classical Timer | This Work Hardware | This Work Software-Like |
|---|---|---|---|---|---|---|---|
| Full TON timer block [ns] | 111.5 | 33.5 | 97.73 | 55.79 | 21.05 | 9 | 9 |
| Coil of TON timer [ns] | 32.75 | 11.5 | n.a. | n.a. | 4.74 | 3 | 3 |
| Read ET cell (MOVE) [ns] | 38.75 | 8 | 9.10 | 4.21 | 13.16 | 3 | 3 |
| Read Q Cell (Switch+Coil) [ns] | 38.75 | 11.5 | 7.27 | 3.68 | 4.74 | 3 | 3 |
| Read ET and Q cells [ns] | 77.50 | 21 | 16.36 | 5.79 | 17.37 | 5 | 5 |
| A Timer.Q [ns] | 19.25 | 9 | 9.15 | 3.89 | 2.42 | 1 | 1 |
| Resolution [ms] | 1 | 1 | 1 | 1 | 10, 100 [ms], 1, 10 [s] | 1 | 1 |

n.a.—non-applicable.

## 6. Conclusions

The design of timer FBs compliant with the IEC 61131-3 standard is proposed in the article. Four types are presented: TON, TOF, TP, and universal. Each type is designed to be fully hardware or software-like. The multi-channel structure is provided.

It is noteworthy that the software-like design is implemented without edge detectors. Such a feature is obtained by reversing the method of time determination by counting the difference between the start and end times and by using specific features of the D flip-flops, that is, clock-enable inputs.

Thanks to the universal design and minimalistic interface, the proposed FBs can be used for hardware support of off-the-shelf units or be integrated into newly built PLC CPUs. The presented timers may be implemented using in FPGAs as well as integrated with the CPU in the form of presented in the paper bit.WORD integrated hardware–software PLC. The paper focuses on the solution of the CPU, which implements the control program in the classical way (i.e., serial–cyclic). The timers themselves are built-in hardware. The task of the program is only to control them appropriately. Therefore, the operation of the timer algorithm itself is entrusted to hardware and is done concurrently to main tasks performed by the CPU.

The presented timers solutions are described in Verilog HDL. They are implemented in an FPGAs with a dedicated software core microprocessor. The paper presents the results of the implementation in an FPGA of the Kintex UltraScale+ family from AMD-Xilinx. Both the classical hardware timer built on the basis of flip-flops and software-like are hardware-based solutions. Software-like timers are based on memory cells on which operations are performed. Both structures are very fast and make it possible to execute the timer update operation in a single clock cycle. However, hardware timers provide faster structures, and they utilize flip-flops, while software-like timers are slower designs but are based on memory-dedicated FPGA structures. The results of the implementation in an FPGA are reported and prove the high efficiency of the proposed solution.

**Author Contributions:** Conceptualization, M.C. and R.C.; Software, R.C.; Validation, A.M.; Investigation, M.C., R.C. and A.M.; Resources, M.C. and R.C.; Writing—original draft, M.C. and R.C.; Writing—review & editing, M.C., R.C. and A.M.; Supervision, M.C. and R.C. All authors have read and agreed to the published version of the manuscript.

**Funding:** This work was supported by the Polish Ministry of Science and Higher Education funding for statutory activities.

**Conflicts of Interest:** The authors declare no conflict of interest.

## Abbreviations

The following abbreviations are used in this manuscript:

| | |
|---|---|
| CPU | Central Processing Unit |
| CR | Current Result |
| ET | Elapsed Time |
| FB | Function Block |
| FPGA | Field Programmable Logic Array |
| IEC | International Electrotechnical Commission |
| IL | Instruction List |
| PII | Process Image Inputs |
| PIQ | Process Image Outputs |
| PLC | Programmable Logic Controller |
| PT | Preset Time |
| SFB | Standard Function Block |
| TOF | Time OFF-Delay Time |
| TON | Time ON-Delay Time |
| TP | Pulse Timer |

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
