# Peer review of "FPGA Implementation of IEC 61131-3-Based Hardware-Aided Timers for Programmable Logic Controllers"

_electronics, doi:10.3390/electronics12204255_

Round 1
Reviewer 1 Report
I would suggest a broader search for related works, considering other authors. This can be useful to improve the justification of the proposal presented.
In general terms, the article presented has a good methodological sequence. One aspect to improve is the introduction. Based on the problems indicated in this part, review the conclusions and add some contributions made that may have been omitted.
On the other hand, include the terms of all the abbreviations that appear in the document, including those that appear in the figures (e.g. Fig, 4. PII, PIQ).
I suggest using different markers (circles, boxes, triangles) in figure 16.
In general terms I find the document well written except for one detail.
Check spelling in figure 18, particularly in the shaded block.
Author Response
Dear Reviewer 1
Comments and Suggestions for Authors and Answers:
1. I would suggest a broader search for related works, considering other authors. This can be useful to improve the justification of the proposal presented.
Authors: We did a broader literature review. We added 10 new literature items (see new References), of course adding relevant descriptions in the Introduction section.
2. In general terms, the article presented has a good methodological sequence. One aspect to improve is the introduction. Based on the problems indicated in this part, review the conclusions and add some contributions made that may have been omitted.
Authors: We did a review of the abstract, contributions and conclusions. According to another reviewer, the conclusions should be rather shortened, so we tried to balance the changes on it.
3. On the other hand, include the terms of all the abbreviations that appear in the document, including those that appear in the figures (e.g. Fig, 4. PII, PIQ).
Authors: Done.
4. I suggest using different markers (circles, boxes, triangles) in figure 16.
Authors: Done
5. Comments on the Quality of English Language.
Authors: We went through the article and corrected typos.
6. Check spelling in figure 18, particularly in the shaded block.
Authors: Done. The typo has been corrected. We have improved the quality of the drawing.
Thanks a lot for Yours comments.
Authors

Reviewer 2 Report
Performance Metrics: Include detailed performance metrics, such as response times and resource utilization, for both hardware and software-like timer designs. This would help readers make informed decisions when choosing a design for their specific applications.
Real-World Testing: Conduct real-world testing of the proposed timers in an industrial setting to validate their functionality, robustness, and suitability for practical PLC applications. Share the results and insights gained from these tests.
Expanded Discussion on FPGA-Based PLCs: Elaborate on the advantages and challenges of using FPGAs as a CPU hardware base for PLCs. Discuss any potential trade-offs and considerations that engineers and system integrators should be aware of when adopting this approach.
The quality of English language in the paper is generally good. The authors demonstrate a strong command of English, with clear and coherent writing. The use of technical terminology and jargon is appropriate and consistent with the subject matter, which is important for a paper of this nature.
Author Response
Dear Reviewer 2
Comments and Suggestions for Authors
1. Performance Metrics: Include detailed performance metrics, such as response times and resource utilization, for both hardware and software-like timer designs. This would help readers make informed decisions when choosing a design for their specific applications.
Authors: Performance metrics covering the area are included in Tables 1 and 2 (for the timer block alone and the block inside the CPU). The speeds of individual solutions (max frequency) are also included in Table 2. Quality metrics covering speed, on the other hand, are included in Tables 3 and 4. In the PLC area, 'execution time' rather than 'response time' is a more appropriate name. But in this case it comes out to one thing.
2. Real-World Testing: Conduct real-world testing of the proposed timers in an industrial setting to validate their functionality, robustness, and suitability for practical PLC applications. Share the results and insights gained from these tests.
Authors: We agree. At this stage, our solution is at the TRL3/4 level. We have laboratory proven the correctness and efficiency of the timers. We are currently working on finalizing the units of (several) CPUs. We hope to be able to test our solution in industry in the future. Industry 4.0 places higher demands on PLCs than has been the case so far.
3. Expanded Discussion on FPGA-Based PLCs: Elaborate on the advantages and challenges of using FPGAs as a CPU hardware base for PLCs. Discuss any potential trade-offs and considerations that engineers and system integrators should be aware of when adopting this approach.
Authors: In the current article, we have expanded the introduction somewhat and added new literature (see References and Introduction parts). Your comment is valuable to us, but we had planned to do such an analysis in an article on our proposed CPUs. There it seems to be much more to the point.
4. Comments on the Quality of English Language
The quality of English language in the paper is generally good. The authors demonstrate a strong command of English, with clear and coherent writing. The use of technical terminology and jargon is appropriate and consistent with the subject matter, which is important for a paper of this nature.
Authors: We have corrected typos.
Thanks a lot for Yours comments.
Authors

Reviewer 3 Report
This research paper introduces three IEC 61131-3 compliant timer function blocks (FBs): timer-on, timer-off, and timer-pulse. These FBs, implemented in Verilog on an FPGA chip (Kintex UltraScale+ family), offer both hardware and software-like designs, with a unique software-like approach that eliminates the need for edge detectors. They can serve as multi-channel timers and are adaptable for use in existing PLCs or as part of new PLC CPUs. Followings are my concerns:
1. The abstract has room for improvement to better emphasize the work accomplished.
2. The introduction section would be enriched by including an exploration of various related techniques. E.g., High Performance FPGA Implementation of Single MAC Adaptive Filter for Independent Component Analysis.
3. The clarity of the contributions presented in this work could be enhanced.
4. The manuscript could benefit from enhancements in its language and overall linguistic quality.
5. The resolution of Figure 18 needs enhancement.
6. The conclusion section appears overly lengthy; kindly consider reducing its size.
Minor editing of English language required
Author Response
Dear Reviewer 3
Comments and Suggestions for Authors
1. The abstract has room for improvement to better emphasize the work accomplished.
Authors: Done. Please see a new vesrion of Abstract.
2. The introduction section would be enriched by including an exploration of various related techniques. E.g., High Performance FPGA Implementation of Single MAC Adaptive Filter for Independent Component Analysis.
Authors: We did a broader literature review. We added 10 literature items (see new vesrion of References), of course adding relevant descriptions in the Introduction section. However, the indicated work seems to be quite far from the topic of the article. Perhaps you have another suggestion as to what we could add to the literature review?
3. The clarity of the contributions presented in this work could be enhanced.
Authors: We agree. Another reviewer also noted that. We have made corrections.
4. The manuscript could benefit from enhancements in its language and overall linguistic quality.
Authors: The article was reviewed by a native-speaker who has a doctorate in a field related to the article's subject matter. We do not feel we can enrich the article ourselves.
5. The resolution of Figure 18 needs enhancement.
Authors: Done. We also corrected a typo in the drawing.
6. The conclusion section appears overly lengthy; kindly consider reducing its size.
Authors: Done. According to another reviewer, the conclusions should be rather expanded, so we tried to balance the changes.
7. Comments on the Quality of English Language. Minor editing of English language required.
Authors: We went through the article and corrected typos. The article was previously checked by a native-speaker expert, who suggested a lot of changes.
Thanks a lot for Yours comments,
Authors

Reviewer 4 Report
Scientific research was conducted methodologically correctly through all phases: description of the state of the art, problems with the limitations of existing software solutions. The solution proposed in the paper is to construct a controller with integrated hardware blocks which support the operations performed in an integrated hardware-software PLC (IHSPLC).
The paper presents 18 pictures, 4 tables and 6 lists of algorithms that enrich the description of the author's research and contribute to the importance and understanding of scientific research.
Experimental results and functional verification have been in this scientific study was performed with software-like and hardware timers were implemented using Verilog HDL.
According to the authors' results, the analog software-like solution provides a 17% lower frequency, which is significant less in relation to the superior hardware solution.
Overall, the research is beautifully and correctly described, and the results are presented graphically and numerically, quantitatively and qualitatively, and proven verified.
The only remark refers to the bibliography, which should be larger, including more available titles on the research topic from open access sites.
Dear Editor-in-chief,
Overall, the research thesis is methodical, structurally and correctly described, and the results are presented graphically and numerically, quantitatively and qualitatively. Also results are proven and verified.
I recommend the publication of this interesting and useful work in your journal.
Kind regards,
Vladimir Tudić, PhD
Author Response
Dear Reviewer 4
Comments and Suggestions for Authors
1. The only remark refers to the bibliography, which should be larger, including more available titles on the research topic from open access sites..
Authors: We did a broader literature review. We added 10 literature items (see new vesrion of References), of course adding relevant descriptions in the Introduction section. However, the indicated work seems to be quite far from the topic of the article. Perhaps you have another suggestion as to what we could add to the literature review?
Thanks a lot for Yours comments,
Authors
